# Cultural differences in healthcare: An investigation using cognitive-affective mapping

Eva Buschmeyer[1], Lasse B. Sander[2], Andrea Kiesel[3], Irina Monno[3], Julius Fenn[3] and Kerstin Spanhel[2] 

[1]University of Freiburg Institute of Psychology, Freiburg im Breisgau, Germany; [2]Institute of Medical Psychology and Medical Sociology, University of Freiburg Faculty of Medicine, Freiburg im Breisgau, Germany and [3]Cognition, Action, and Sustainability Unit, University of Freiburg Institute of Psychology, Freiburg im Breisgau, Germany

## Research Article

**Keywords:**
refugees; healthcare facilitators; healthcare barriers; culturally sensitive treatment; mental health needs

**Corresponding author:**
Kerstin Spanhel;
Email: kerstin.spanhel@mps.uni-freiburg.de

## Abstract

Refugees in Germany often do not receive the necessary mental healthcare. Understanding refugees' healthcare needs, that is factors that support or hinder their health, can foster the development of effective, culturally appropriate and accessible health services. The present exploratory study investigated these needs using cognitive-affective maps (CAM), a recently developed mind map-like measurement technique enabling participants to visualise their thoughts and emotions as networks. Thirty refugees from diverse regions of origin (West Africa, Middle East and Ukraine) were asked to indicate factors supporting and hindering their health by drawing a CAM using a digital tool. We analysed the drawn CAM concepts qualitatively according to a five-step procedure, including deductive and inductive category formation. Drawn CAM concepts concerning supporting factors were grouped along the categories 'resources' and 'possibilities for explicit treatments of healthcare needs', suggesting an openness to promote well-being among refugees. Yet, various concepts were grouped along the categories 'psychological challenges', 'specific living conditions in Germany' and 'access to healthcare', focusing on factors hindering well-being. Frequency analyses of these categories suggested differences between the subsamples, which should be further investigated in future studies in order to integrate a possible heterogeneity within refugee populations in offered healthcare.

## Impact statement

Refugees worldwide face a critical gap in access to mental healthcare, despite being disproportionately affected by trauma, stress and displacement-related challenges. This study contributes to closing that gap by introducing an innovative, visual method – cognitive-affective maps (CAM) – to better understand the mental health needs of refugees in Germany. Unlike conventional surveys or interviews, CAMs allow individuals to express both their thoughts and emotions in a mind map-like format, making it accessible across language and cultural barriers.

The findings reveal not only substantial barriers to healthcare access but also an openness among the participating refugees to engage in maintaining their mental and physical health. By highlighting differences in needs and challenges across three regions of origin (West Africa, Middle East and Ukraine), the study underscores the importance of culturally sensitive and tailored healthcare strategies rather than generalised approaches.

Beyond its relevance to researching and improving refugee healthcare, this research demonstrates the broader applicability of CAMs as a low-threshold, adaptable tool for use in diverse socio-cultural contexts. The method can give refugees a direct voice in shaping health research, thereby helping to develop services that are better targeted to the needs of marginalised groups, which are often underrepresented in research. In doing so, the study offers a scalable pathway towards more inclusive healthcare systems – locally in Germany, and internationally wherever forced migration intersects with unmet mental health needs.





## Introduction

By the end of 2024, 36.8 million individuals worldwide were forcibly displaced from their home countries due to persecution, conflict, violence, human rights violations and incidents that significantly disrupt public order (UNHCR, 2025). These individuals, hereafter referred to as refugees, endure multiple stressors before, during and after migration (Steel et al., 2017; Müller et al., 2018), contributing to a high prevalence of mental disorders (Nesterko et al., 2020; Hoell et al., 2021). Germany hosted 2.7 million refugees in 2024, making it the fourth-largest host country of refugees worldwide (UNHCR, 2025).

Besides pre-migratory trauma and the migration process itself, post-migratory living conditions crucially influence the mental health of refugees. Nutsch and Bozorgmehr (2020) highlight that *security of residence status*, *satisfaction with accommodation* and opportunities for *social and economic participation* considerably impact on refugees' mental health. However, despite the universal "right to a standard of living adequate for the health and well-being of himself and of his family" being stated in Article 25 of the Universal Declaration of Human Rights (United Nations General Assembly, 1948), only an estimated 5% of refugees with mental disorders receive the necessary mental healthcare in Germany (BAfF, 2016). In contrast, around 19% of the German population in need of mental healthcare receive corresponding treatment (Mack et al., 2014). This underscores a critical mental health treatment gap among refugees (Nowak et al., 2022).

Access barriers are widely recognised as a primary cause of this gap (Bozorgmehr and Razum, 2015). Bozorgmehr and Gold (2023) categorised these barriers into three types: *generic barriers* (which can affect any patient and are not related to migration or refugee identity, for example health literacy or health-seeking behaviour), *migration-specific barriers* (which can also occur among other migrant populations like international students or international workers, e.g. language barriers or lack of procedural knowledge) and *refugee-specific barriers* (which occur due to the specific legal situation of refugees, e.g. issues related to housing or relocations within the host country). Notably, the legal status of refugees in Germany is not uniform and depends on the region of origin, reasons for flight and the ability to substantiate these reasons (Federal Office for Migration and Refugees, 2024). Differences in status lead to substantial disparities in living conditions and healthcare access. For instance, refugees often fall under the Asylum Seekers' Benefits Act (in German: Asylbewerberleistungsgesetz, AsylbLG) for the first 36 months after their arrival – limiting access to the treatment of mental and chronic diseases (Biddle, 2024). In contrast, Ukrainian refugees, who are granted protection under the European Union's Temporary Protection Directive, benefit from immediate access to the German housing and labour market, as well as education and healthcare (Heiermann and Atanisev, 2024).

Such a possibility of direct access to professional healthcare for Ukrainian refugees in Germany is essential: adequate professional healthcare is a fundamental factor for the promotion of mental health among refugees (Schlechter et al., 2021). In addition, other factors also play an important role in supporting the mental health of refugees (Walther et al., 2021). These factors encompass personal resources, such as *available resources*, *coping strategies* and *social support*, which can help deal with burdens and reduce stressors specific to the living situation (Nowak, 2022). Tailoring these supporting factors to the individual refugees' needs seems crucial (Lebano et al., 2020). However, knowledge about refugees' health needs remains limited, particularly from their own perspectives and considering potential differences based on their regions of origin (Frank et al., 2017; Lebano et al., 2020; Lueckenbach, 2023).

The present study aims to address this research gap by exploring the self-perceived health behaviour and needs of refugees in Germany using cognitive-affective maps (CAM; Reuter et al., 2022; Thagard, 2010; Thagard and Larocque, 2020), a recently developed, mind map-like measurement technique. We assess individual beliefs, experiences and emotions with CAMs based on Thagard's HOTCO ("hot coherence") model, which provides a theoretical foundation for understanding decision-making processes by integrating emotions into cognitive evaluations. Unlike purely rational models, HOTCO conceptualises decision-making as an interaction between beliefs, values and emotional valences, where coherence refers to the extent to which these elements fit together in a structured network (Thagard, 2000, 2006, 2010). Within this framework, CAMs serve as cognitive-affective networks that visually represent individuals' efforts to achieve coherence in their understanding of healthcare-related factors (Thagard and Larocque, 2020). The structured nature of CAMs implies that concepts are added or adjusted in a way that enhances internal consistency, thereby reflecting underlying cognitive and emotional patterns (Johnson-Laird, 1983; Denzau and North, 1994). Taking advantage of these characteristics, studies have been conducted with CAMs using qualitative (Clapp, 2021), quantitative (Mansell et al., 2021) and combined methods (Fenn et al., 2023). CAMs have also been used in various research contexts, for example to investigate political ideologies (Thagard, 2015) or attitudes (Reuter et al., 2021), to contribute to conflict resolution (Thagard, 2021), or to identify ethical principles (Höfele et al., 2024). In addition, there was also examined the success of an early childhood education in one study (Luthardt et al., 2020). CAMs were initially mostly drawn by researchers to illustrate and visualise existing data material (Homer-Dixon et al., 2013; Findlay and Thagard, 2014; Luthardt et al., 2020). Yet, recently they have also been used as a method to directly assess study participants' opinions, by having CAMs been drawn by the participants to express their own perspectives (Reuter et al., 2022; Fenn et al., 2023). Also, a first study revealed reasonable retest-reliability for quantitative and qualitative data (Gros et al., 2024).

Applying this perspective to healthcare, CAMs allow the identification of shared and distinct healthcare concerns among refugee subgroups. This study focuses on identifying supporting and hindering factors experienced by refugees in the context of both mental and physical health. This goes back to the fact that many cultures do not strictly separate body and mind (Golsabahi-Broclawski et al., 2020). As Homer-Dixon et al. (2013) suggested, CAMs function as cognitive-affective networks, revealing underlying belief structures, tensions and synergies within different communities. By including refugees from diverse regions of origin (West Africa, Middle East and Ukraine), the study acknowledges and explores potential differences in health beliefs and healthcare needs (Yilmaz-Aslan et al., 2018; Golsabahi-Broclawski et al., 2020). These regions of origin were selected based on the fact that the Middle East and Ukraine currently have the largest refugee populations in the world in need of international protection (UNHCR, 2025). Refugees from West Africa are included to create a comparison group with a further cultural background. Given that refugees from West Africa, the Middle East and Ukraine may have diverse cultural perceptions of health and illness, CAMs could effectively capture these regional differences while also identifying commonalities (Thagard and Larocque, 2020). Moreover, the emotional valence encoded in CAMs provides insights beyond traditional surveys, capturing the implicit attitudes and affective significance that refugees assign to different healthcare factors (Homer-Dixon et al., 2013; Livanec et al., 2022). This could be crucial for designing culturally adapted interventions that align with refugees' lived experiences. Hence, the following research questions are posed: 1) What supporting and hindering factors do refugees experience in healthcare? And 2) What differences or commonalities are there between refugees from different regions of origin?

## Methods

### Recruitment and participants

The Ethics Committee of the University of Freiburg in Germany granted ethical approval for the study (application number: 23–1548-S2). In March 2024, a convenience sample of $N = 32$ refugees was recruited through personal contacts and offers of help organisations for refugees in and around Freiburg, Germany. To this end, one researcher (EB) personally visited various locations (e.g. language courses, sports activities, Ukrainian culture centre, refugee accommodation in Freiburg, Germany), directly addressed refugees to inform them about a possible study participation and, if feasible, to conduct it. In some cases, study participants told their family and friends about the study, which helped bring in new participants. Inclusion criteria were a minimum age of 18 years, sufficient language skills in German or English or the possibility of interpretation, flight background and being originally from West Africa, Middle East or Ukraine. Participants from the individual countries of origin were grouped into regions in order to include persons from as many different countries of origin as possible, given the small number of participants, while still providing a basis for comparisons. The participation was not financially compensated, but five 20€ vouchers were raffled among the participants. Of the 32 people who started the study, one person withdrew during the study due to language difficulties, and another person withdrew due to other time commitments. The final sample consisted of $N = 30$ participants. The sample size is based on the exploratory and qualitative character of the study investigating CAMs in a refugee population, but also allows insights into possible differences depending on the region of origin.

Of the 30 participants, 60% were 18–34 years old, 30% were 35–49 years old and 10% were 50 years or older. About half of them (54%) were female. Eleven participants came from West Africa (seven from Gambia, two from Guinea, one from Nigeria and one from Cameroon), 10 from Middle East (five from Syria, four from Afghanistan and one from Iran) and nine from Ukraine. One participant from Gambia had never used health services in Germany before. Differences between the three subsamples were shown in age, gender, residence status, availability of health insurance, occupation, reasons for flight, average time in Germany, education and self-assessed socio-economic status (see Table 1).

### CAM measurement technique

CAMs are a recently developed mind map-like measurement technique that can be applied to visualise individual beliefs, experiences and emotions (Fenn et al., 2025). CAMs provide an innovative alternative to traditional questionnaires and interviews to explore complex psychological constructs in an engaging and interactive way (Thagard, 2010, 2012; Thagard and Larocque, 2020; Livanec et al., 2022; Reuter et al., 2022). Figure 1 shows an exemplary CAM. As outlined in Thagard (2010), CAMS are a network of concept nodes and lines linking the concept nodes. The concepts can denote any content in text form, such as thoughts, emotions, events or factual knowledge. Each concept conveys an affective valence, depicted by the colour and shape of the concept node. In the present study, three colours and shapes were used: green ovals indicate positive valence, red hexagons indicate negative valence; yellow boxes indicate neutral valence (both positive and negative affect or neither of both). The lines connecting the concepts are either solid, indicating a positive link (reinforcement/support), or dashed, indicating a negative link (inhibition/opposition). Positive and negative concepts, as well as solid and dashed lines, have three levels of intensity (+1 to +3 or −1 to −3 respectively), represented by the thickness of the concepts' frame or line. The CAMs were drawn and analysed with the

**Table 1.** Demographics and socio-cultural characteristics of the sample of refugees participating in the study in Germany

| Variable | Total sample ($N = 30$) | Region of origin | | |
| --- | --- | --- | --- | --- |
| | | West Africa ($n = 11$) | Middle East ($n = 10$) | Ukraine ($n = 9$) |
| Gender (% Female) | 53 | 36 | 50 | 78 |
| Country of origin | | Cameroon (1), Gambia (7), Guinea (2), Nigeria (1) | Afghanistan (4), Iran (1), Syria (5) | Ukraine (9) |
| Age: % < 35 years old | 60 | 82 | 70 | 22 |
| Residence status | | | | |
| % German nationality/permanent | 23 | 9 | 60 | 0 |
| % Temporary | 57 | 36 | 40 | 100 |
| % None/ tolerated | 17 | 45 | 0 | 0 |
| Health service usage (% at least once) | 97 | 91 | 100 | 100 |
| Health insurance (% available) | 77 | 36 | 100 | 100 |
| Education (% with University degree) | 47 | 9 | 50 | 89 |
| Occupation | | | | |
| % Student (school, university, German classes) | 37 | 36 | 50 | 22 |
| % Employment | 23 | 18 | 10 | 44 |
| % No employment | 40 | 45 | 40 | 33 |
| Months in Germany, M (SD) | 60.4 (45.2) | 87.4 (50.4) | 69.3 (41.2) | 23.4 (1.7) |

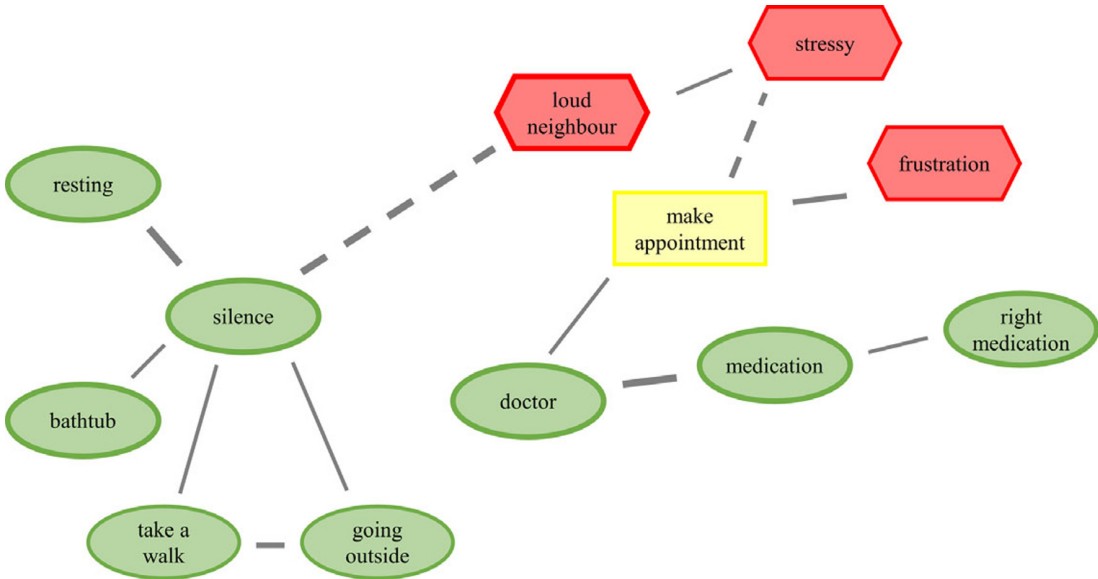

**Figure 1.** Exemplary Cognitive-Affective Map (CAM) drawn by a refugee from Gambia participating in the exploratory study in Germany. The task was to think of self-perceived supporting and hindering factors to improve the own well-being when feeling unwell. Properties of the CAM: positive (green), negative (red) and neutral (yellow) concepts; reinforcing concepts (solid lines), oppositional concepts (dashed lines), stronger relationships (thicker lines).

C.A.M.E.L. software (Fenn et al., 2025) and stored on a local JATOS server (version 3.8.5; Lange et al., 2015).

### *Procedure*

The study material was offered online and the participants could either conduct the study independently (*n* = 6; 1 participant from the West African subsample, three participants from the Middle Eastern subsample and two participants from the Ukrainian subsample) or together with one researcher (EB; *n* = 24), if difficulties or concerns arose. They could also choose whether they wanted to conduct the study in German or English language. Nineteen participants conducted the study in German (two participants from the West African subsample, nine participants from the Middle Eastern subsample and eight participants from the Ukrainian subsample), 11 participants conducted the study in English (nine participants from the West African subsample, one participant from the Middle Eastern subsample and one participant from the Ukrainian subsample). In three cases of joint study conduction, the study participation was enabled through translation by third parties (two Persian and one Russian translation). The study lasted between 30 min (independent study conduction) and 1 hour (joint study conduction). In the joint study

conduction, the researcher read aloud all instructions and drew the answers to the posed questions in the CAM format. All data were linked with an automatically created study ID.

After reading the study and data protection information either independently or with additional verbal communication, the participants provided digital informed consent to participate in the study. The participants were then instructed on how to draw a CAM with the online application through written online instructions, supplemented by verbal instructions where necessary. Following, the participants were asked to draw a CAM considering two questions (see Figure 2). Based on the feedback gained in a pilot study with nine participants (of these, three as representatives of the target group), the wording of the question was developed in an iterative process in relation to the aim of our study. The CAMs were drawn in a free-response format; no predefined concepts were given. The participants had to draw at least eight concept nodes and lines.

After drawing their CAM, the participants were forwarded to SoSci Survey (Leiner, 2024), where they rated "how well [they] could describe in the mind map which help [they] need/ what could make it difficult to get help when [they] are not feeling well" (two scales from 1 = *totally unrepresented* to 7 = *totally represented*). Furthermore, they were asked socio-demographic questions (e.g. gender,

Imagine your life in Germany right now at this moment. Now imagine this: You are **not well**. For example, because you are physically ill or because you are mentally unwell. Draw the mind map for two questions:
*(Click on the questions. Then you get more ideas in which direction you could answer the questions.)*

**1. Which kind of help would you need to feel better?**
 Possible ideas to answer the question:
 What can you do for yourself, for example?
 What can other people do for you?
 What circumstances can help you in this situation?

**2. What could make it difficult to get the help you need?**
 Possible ideas to answer the question:
 What is difficult for you in the situation?
 Who could make the situation more difficult?
 What factors could make it difficult to implement things of question 1?

**Figure 2.** The task given to the refugees participating in the study in Germany to draw cognitive-affective maps.

age, residence status) based on various questionnaires (Adler et al., 2000; DIW Berlin/SOEP, 2016; Bartig et al., 2023). After completing the questions, the participants could sign up for the voucher raffle.

## Data analysis

Descriptive analysis of the socio-demographic data was conducted using SPSS Statistics version 29 (IBM Corp, 2023). The preparation of the raw CAM data (translating German CAMs to English, matching CAM ID with socio-demographic data) was conducted using R Statistical Software version 4.3.1 (R Core Team, 2023). The summary and analysis of the CAMs were conducted by using the open-source C.A.M.E.L. software (Fenn et al., 2025). A descriptive analysis of the CAMs was conducted, reporting the mean number of drawn concepts and their mean valence (negative concepts = −1; positive concepts = +1 and neutral concepts = 0) separately for the three subsamples.

For the qualitative analysis of the CAM data, synonymous concepts were first identified and merged into overarching terms. To then summarise the qualitative CAM data, a modified version of the five-step procedure by Fenn et al. (2023) was applied by one researcher (EB) who developed a coding guideline (see Supplementary Tables): (1) Main categories and subcategories were extracted from the existing literature in a deductive process, whereby three category systems were built: one for supporting factors, one for hindering factors and one for descriptive concepts. The concepts in the CAMs were classified into the extracted categories based on the content and the valence that the participants ascribed to the concepts. (2) For the subsequent inductive step, concepts that could not be assigned to one of the deductively derived main categories or subcategories were examined for similarities in content, and additional main categories and subcategories were formed inductively. Additionally, dimensions (sub-subordinate categories) were inductively determined from the material. (3) The processes (1) and (2) were repeated iteratively until all concepts drawn in the CAMs were coded. It should be noted that neutral concepts could be categorised simultaneously in both the supporting and hindering factors categories, whereas positive and negative concepts could only be categorised once. Uncertainties during the process of (3) were resolved in discussion with the researcher KS. (4) The category systems were reduced by merging dimensions into the next higher subcategories. (5) To enable the CAM data to be described quantitatively, too, the absolute frequencies for each subcategory per CAM (i.e. per person) were calculated. From this, the mean frequency and its standard error of the mean were calculated for each subcategory per CAM, both for the total sample and for each of the subsamples (West Africa, Middle East and Ukraine). The frequency thus provides information on how many concepts from the respective subcategory were drawn on average per CAM (i.e. per person) and is a number larger than zero. Due to the small number of participants per group, the frequencies are presented descriptively and we refrain from statistical analyses.

To assess the interrater reliability of the coding guideline, Cohen's kappa was calculated to determine the agreement between two raters for classifying concepts into the designated subcategories. This assessment was conducted on 10% of the data material. Kappa was found to be $\kappa = 0.90$, indicating "almost perfect" agreement according to the benchmarks of Landis and Koch (1977).

## Results

### Descriptive analysis of CAMs

The 30 CAMs contained 499 concepts in total. Each of the 30 CAMs included on average 16.63 ($SD = 6.14$) concepts, of which 45% were positive, 39% negative and 16% neutral. The mean valence for all concepts was 0.19 ($SE = 0.11$). Participants from the West African subsample drew on average 13.45 ($SD = 3.53$) concepts with a mean valence of 0.21 ($SE = 0.21$), participants from the Middle Eastern subsample drew on average 16.60 ($SD = 6.36$) concepts with a mean valence of $-0.09$ ($SE = 0.18$) and participants from the Ukrainian subsample drew on average 20.56 ($SD = 6.65$) concepts with a mean valence of 0.43 ($SE = 0.17$).

### Category system of supporting factors

The category system of supporting factors included two main categories, five subcategories, 25 dimensions and 258 concepts (179 without synonyms). The main categories and the subcategories with their mean frequencies of concepts and exemplary concepts, occasionally with comments, are illustrated below. Additionally, the dimensions and all concepts are shown in Supplementary Table S1.

The main category *resources* included 141 concepts related to dealing with stress and to reducing stressors (indirect disease management); it comprised three subcategories. The subcategory *available resources* ($M = 0.70$, $SE = 0.23$) included internal factors available to the individual person to cope with stressful situations, for example "independence", "good work" and "learning German." The concept "many minijobs currently" was commented on by a participant from Iran: "hard work, does not match actual professional qualifications". The subcategory *coping strategies* ($M = 2.57$, $SE = 0.40$) included daily activity strategies that help individuals to improve their well-being, such as "sport", "praying" and "outside." A participant from Guinea commented on their concept "listening to music" that this "makes you thinking far away, gives hope". The subcategory *social support* ($M = 1.43$, $SE = 0.27$) involved support from personal relationships, such as "ask my family", "meeting friends" and "support in everyday life". The concept "nice people" was drawn by a participant from Ukraine, and the person commented that this "is in the centre of Germany for me; Germany is very positive for me".

The main category of *explicit treatments of healthcare needs* included 117 concepts related to the uptake of healthcare (direct disease management); it comprised two subcategories. The subcategory *professional treatment* ($M = 3.37$, $SE = 0.40$) referred to strategies that require the involvement of healthcare professionals, for example "contact social worker", "health insurance", "psychotherapy" and "correct diagnosis." A participant from Ukraine noted on their concept "English skills in the health system" that "it is very helpful if someone can speak English". The subcategory *self-administered treatment* ($M = 0.53$, $SE = 0.16$) included strategies that the participants can implement on their own, for instance "sleeping" (a participant from Nigeria commented "take a rest") and "drinking water".

Figure 3 shows the mean frequencies of concepts per subcategory of supporting factors for the total sample and separately for the subsamples. Visual inspection of the mean frequencies reveals that *professional treatment* was the subcategory with the most coded concepts, with $M = 3.37$ ($SE = 0.40$) concepts drawn per participant, followed by the subcategory *coping strategies*, with $M = 2.57$ ($SE = 0.40$) concepts drawn per participant. Concepts of the main category *explicit treatments of healthcare needs* were mentioned particularly often by the participants from the Ukrainian subsample ($M = 6.44$, $SE = 0.92$ concepts per participant).

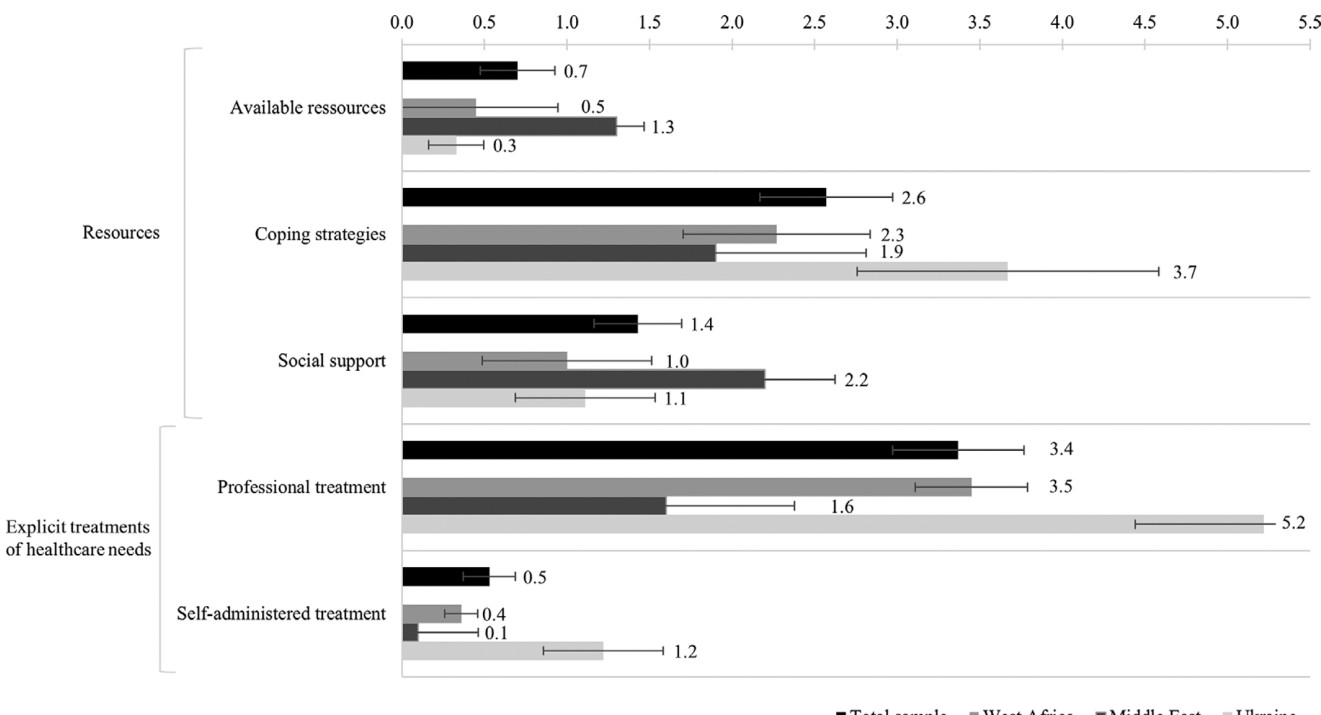

**Figure 3.** Mean frequencies of concepts mentioned per subcategory of supporting factors per cognitive-affective map drawn by the refugees participating in the study in Germany; illustrated for the total sample ($N = 30$) and for each region of origin separately (West Africa, $n = 11$; Middle East, $n = 10$ and Ukraine, $n = 9$). The error bars show the standard error of the mean.

In contrast, the participants from the Middle Eastern subsample, across the various countries of origin, rather named concepts of the main category *resources* ($M = 5.40$, $SE = 0.73$ concepts per participant), and the concepts of the participants from the West African subsample, across the various countries of origin, were distributed quite evenly across the two main categories (*resources*: $M = 3.73$, $SE = 0.62$ concepts per participant; *explicit treatment of healthcare needs*: $M = 3.82$, $SE = 0.44$ concepts per participant).

### Category system of hindering factors

The category system of hindering factors included four main categories, 10 subcategories, 16 dimensions and 207 concepts (141 without synonyms). The main categories and the subcategories with their mean frequencies of concepts and exemplary concepts are illustrated below. Additionally, the dimensions and all concepts are shown in Supplementary Table S2.

The main category *access to healthcare* included 109 concepts that described the possibility of making use of general healthcare in Germany; it comprised three subcategories. The subcategory *generic access barriers* ($M = 1.63$, $SE = 0.46$) involved barriers that can affect anyone, such as "long waiting time" (a participant from Cameroon commented "to get a doctor appointment"), "sometimes rude" and "financially not possible". The subcategory *migration-specific access barriers* ($M = 1.17$, $SE = 0.31$) included barriers that all migrant populations face, including international students or workers, such as "language barrier", "people not understanding" and "no information health system". For example a participant from Ukraine drew the concept "German healthcare system complicated" and commented: "I don't know how it works; where to get emergency telephone numbers; which number to call if I have a leg injury and a head injury, or is it the same number?" The subcategory *refugee-specific access barriers* ($M = 0.83$, $SE = 0.20$) consisted of barriers specific to refugees, such as "no health insurance", "bureaucracy"

and "unclear responsibilities". A participant from Syria commented on their concept "migration background": "I can't cope with all the structures, formalities and deadlines."

The main category *living conditions in Germany* comprised 24 concepts concerning the everyday life, described in four subcategories: *residence status* ($M = 0.10$, $SE = 0.06$), for example "fear of court"; *satisfaction with accommodation* ($M = 0.13$, $SE = 0.06$), for example "little bed camp" (a participant from Cameroon commented "makes pain, sometimes falling out of bed because it is too small"); *social participation* ($M = 0.27$, $SE = 0.08$), for example "lack of social support" (a participant from Syria commented "homelessness, unemployment, family problems") and "no help"; and *economic participation* ($M = 0.30$, $SE = 0.11$), for example "no work permit."

The main category of *psychological challenges* involved 69 concepts that addressed conditions that negatively affect psychological well-being; it comprised three subcategories. The subcategory *generic psychological challenges* ($M = 1.67$, $SE = 0.32$) involved psychological challenges that can affect anyone, such as "conflict in relationship", "not feeling good" and "sleeping difficulties". The concept "stay home" was commented on by a participant from Gambia: "be myself, cook something; you can't stay home all the time". The subcategory *migration-specific psychological challenges* ($M = 0.33$, $SE = 0.16$) included psychological challenges that all migrant populations might face, such as "cultural differences" and "homesickness". A participant from Afghanistan, for example, drew a concept "no family communication" and commented that "due to Afghan culture, I cannot discuss mental health issues with my family". The subcategory *refugee-specific psychological challenges* ($M = 0.30$, $SE = 0.12$) consisted of psychological challenges specific to refugees, such as "growing up in war" and "not doing anything [due to no work permission]". The same participant from Afghanistan commented on their concept "traumas" the associated "flashbacks" experienced.

The fourth main category, *other hindering factors* ($M = 0.17$, $SE = 0.07$), included 5 remaining concepts that were considered to

belong to "hindering factors" but did not fit into any other category of this category system, for example "if it doesn't help" or "somebody disturbing me."

Figure 4 shows the mean frequencies of concepts per subcategory of hindering factors for the total sample and separately for the subsamples. Visual inspection of the mean frequencies reveals that *generic psychological challenges* and *generic access barriers* were the subcategories with the most coded concepts, with $M = 1.67$ ($SE = 0.32$) concepts per participant drawn concerning *generic psychological challenges*, and $M = 1.63$ ($SE = 0.46$) concepts per

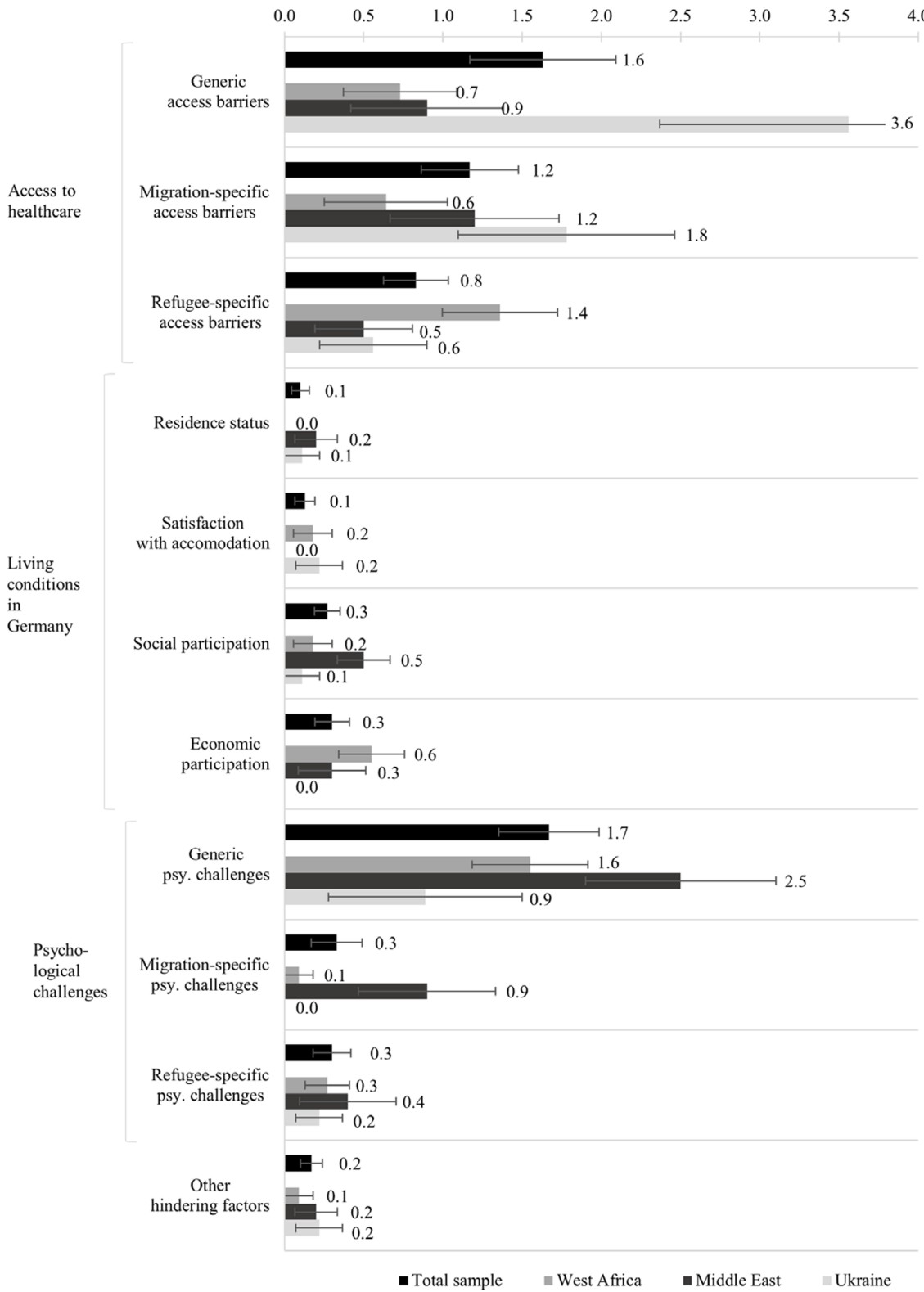

**Figure 4.** Mean frequencies of concepts mentioned per subcategory of hindering factors per cognitive-affective map drawn by the refugees participating in the study in Germany; illustrated for the total sample ($N = 30$) and for each region of origin separately (West Africa, $n = 11$; Middle East, $n = 10$ and Ukraine, $n = 9$). The error bars show the standard error of the mean.

participant drawn concerning *generic access barriers.* Concepts of the main category *living conditions in the host country* were infrequently mentioned by all subsamples ($M = 0.80$, $SE = 0.17$ concepts per participant). The participants from the Ukrainian subsample particularly often mentioned concepts of the subcategory *generic access barriers* ($M = 3.56$, $SE = 1.20$ concepts per participant) and *migration specific access barriers* ($M = 1.78$, $SE = 0.68$ concepts per participant), whereas the participants from the West African subsample across the various countries of origin rather named *refugee-specific access barriers* ($M = 1.36$, $SE = 0.36$ concepts per participant). On average, the participants from the Middle Eastern subsample, across the various countries of origin, mentioned many *psychological challenges* ($M = 3.80$, $SE = 1.00$ concepts per participant), whereas the participants from the Ukrainian subsample mentioned few ($M = 1.11$, $SE = 0.59$ concepts per participant). More *generic psychological challenges* were named ($M = 1.67$, $SE = 0.32$ concepts per participant) as compared to *migration-specific psychological challenges* ($M = 0.33$, $SE = 0.16$ concepts per participant) or *refugee-specific psychological challenges* ($M = 0.30$, $SE = 0.12$ concepts per participant).

### Exploratory findings

Five participants ($n = 5$) expressed their gratitude for taking part in the study in written feedback. They were of different ages and came from the three different regions of origin represented in the study. Each of them showed below-average satisfaction with health as compared to the overall sample. For example a female participant from Iran wrote: "It helps to talk about it. The mind map helped to talk about it. It can reduce the stress and depression a bit." A male participant from Afghanistan wrote: "Thank you, because I can talk about my difficulties, it takes away some of the stress." A female participant from Cameroon wrote: "Mind map helps to open up." We take this feedback, together with previous positive experiences with the CAM methodology (Livanec et al., 2022) and the researcher (EB)'s subjective positive impression during study conduction, as cautious hints that the mere drawing of a CAM might trigger a process to reflect the situation. Yet, future studies are needed to evaluate whether CAMs might be a suitable instrument to serve as a supporting factor themselves, for example in a psychotherapeutic process.

### Discussion

This study explored the health needs of refugees from diverse regions of origin (West Africa, Middle East and Ukraine), emphasising the importance of considering individual perspectives and contexts. Using a novel CAM measurement technique, the participants identified various psychological challenges that contribute to increased (mental) healthcare needs, and at the same time highlighted supporting factors that indicate an openness to promoting mental and physical well-being. However, several factors that hinder access to healthcare were also reported, underscoring the need to address these barriers. Many aspects were mentioned that correspond with previous literature (Satinsky et al., 2019; Hajak et al., 2021; Childress et al., 2025), indicating that the CAM measurement technique offers good potential for application.

Consistent with the high mental burden among refugees (Blackmore et al., 2020), the participants referred to various psychological challenges. They most frequently mentioned generic psychological challenges (e.g. family or partnership issues), with fewer migration-specific (e.g. distance to home country) and refugee-specific (e.g. trauma) psychological challenges reported. The mentioned psychological challenges faced by the participating refugees can be related to existing research in this field (Scharpf et al., 2021; Kronick et al., 2023). The participants from the Middle Eastern subsample named more (especially generic) challenges than the participants from the West African subsample, who in turn named more challenges than the the participants from the Ukrainian subsample. This aligns with the literature highlighting the impact of post-migration stressors on refugees' mental health (Boettche et al., 2016; Drescher et al., 2021), which may in some cases manifest as generic (in contrast to migration- or refugee-specific) psychological challenges in our categorisation. Accordingly, the fewer challenges reported by the participants from the Ukrainian subsample are potentially attributable to fewer experienced post-migration stressors (MIDEM, 2022).

The participants also identified numerous supporting factors that could help them cope with the experienced psychological challenges. Among these were self-administered strategies as well as professional treatment. Self-administered strategies included social support and coping strategies, for example religious or spiritual activities, consistent with previous literature (Mehjabeen et al., 2023; Childress et al., 2025). This suggests that refugees possess valuable internal resources that can help them cope with the psychological challenges. A resource-oriented perspective in research and healthcare, including the promotion of supporting factors, could empower refugees by promoting active agency and dealing with health-related life crises. The reference to professional treatments in the CAMs underlines the need for mental healthcare among refugees shown in the literature (Satinsky et al., 2019), with the participants from the Ukrainian subsample more frequently citing professional treatment as a supporting factor compared to the participants from the West African and Middle Eastern subsamples. Differences in the healthcare systems in their countries of origin, as well as in their illness perception and understanding, could play a role in the fact that the participants from the West African and the Middle Eastern subsamples named professional treatment options less frequently as supporting factors than the participants from the Ukrainian subsample (Yilmaz-Aslan et al., 2018; Tillmann and Shah Hussaini, 2022). Furthermore, the difference could be related to the structural differences in healthcare provision between Ukrainian refugees, who are granted direct access to healthcare under the European Union's Temporary Protection Directive (Heiermann and Atanisev, 2024), and refugees from other countries of origin, who often have limited access to healthcare under the Asylum Seekers Benefits Act for the first 36 months after arriving in Germany (Biddle, 2024).

The underutilisation of mental healthcare among refugees is also due to various barriers that hinder accessing the supporting factors named by the participating refugees and thus impede their considerable impact. In accordance with the taxonomy of Bozorgmehr and Gold (2023), generic, migration-specific and refugee-specific access barriers were named. The frequency of the barriers mentioned differed between the subgroups and potentially reflects the structural differences between refugees from the different regions of origin described above. The participants from the Ukrainian subsample and, to a lesser extent, from the Middle Eastern subsample mainly mentioned generic barriers, such as aspects of the shortage of doctors or negative professional-patient interaction, as well as migration-specific barriers, such as language barrier-related aspects or a lack of knowledge about the healthcare system (Kiselev et al., 2020; Davitian et al., 2024; Childress et al.,

2025; Kardas et al., 2025). The participants from the West African subsample named more refugee-specific access barriers, such as a lack of legal status and health insurance. This highlights the interplay of socio-legal factors with access barriers, as described by Nutsch and Bozorgmehr (2020), with differences in the legal status of the participants from the Ukrainian and West African subsamples (Nutsch and Bozorgmehr, 2020; MIDEM, 2022): While all participants from the Ukrainian subsample had temporary residence status and health insurance, almost half of the participants from the West African subsample had no residence status, and almost two thirds of them were without health insurance. Thus, participants from the West African subsample may have difficulties accessing healthcare even before they are confronted with generic or migration-specific barriers to access.

All in all, these findings highlight the importance of accessible and appropriate healthcare to overcome the underutilisation of mental healthcare among refugees, including the access to comprehensible information on the healthcare system and on health or illness themselves (Alvarez et al., 2022; Dumke et al., 2024).

Differences in the perceived supporting and hindering factors across the regional groups (West Africa, Middle East and Ukraine) were descriptively revealed, indicating a heterogeneity of refugees in resources, health problems and needs (Frank et al., 2017). Yet, it is important to note that the three groups in our study not only differ depending on the region of origin, but also regarding other socio-cultural and structural factors, such as age, gender, education, employment, residence status, time in Germany, reasons for flight and health insurance access. Although this is in line with existing literature (Brücker et al., 2017, 2023), it must inform the interpretation of group differences.

The present study has several limitations. First, the participants frequently needed help during study conduction due to language barriers and difficulties in drawing CAMs on their own, contrasting other CAM studies (Reuter et al., 2022). This made a joint study conduction necessary in many cases and may have discouraged participants from addressing stigmatised topics, introducing a risk of reactivity such as an inhibition to talk about refugee- or migration-specific psychological challenges. In two cases involving translators with personal relationships to the participants, this risk may have been amplified. Second, the sample was small but heterogeneous. Hence, the descriptively observed differences between the subsamples can only be taken as indications of possible existing differences between various refugee groups. This applies in particular to the individual countries of origin, as the number of people from each country was too small to allow for comparisons, which prevented an in-depth consideration of individual countries and possible differences between them. The risk of homogenising people from different countries of origin under a broad regional term is therefore increased. Third, the small, city-specific sample limits the generalisability of the findings. Fourth, to simplify the study conduction when drawing a CAM, no distinction was made between ambivalent and neutral concepts, and the concepts were limited to three written words. When categorising the concepts, the neutrally marked concepts were checked for ambivalence, which limits the validity of these categorisations. The word limit prevented an in-depth analysis of the content provided by the participants.

The limitations described could be addressed in a future, larger-scale online study, extended to refugees throughout Germany. As far as our exploratory study and previous literature (Livanec et al., 2022) indicate, CAMs appear to be an adequate measurement technique to enable such an investigation. Difficulties refugees faced in the present study to draw CAMs on their own can be attributed to two aspects, which should be considered in future studies. On the one hand, the CAM methodology and explanation have not been adapted to the socio-cultural backgrounds of the participants (e.g. language, illustrations and explanations), which is shown to be essential (Spanhel et al., 2021; Whitehead et al., 2023). On the other hand, personal contact has been found necessary in other studies using digital tools, for example revealing that recruitment was particularly successful via personal contacts (Roehr et al., 2019), or pointing to the importance of feeling seen and being able to establish a trusting relationship with a person (Lindegaard et al., 2021). Having this in mind, a further simplified, visualised, adapted and translated explanation of the CAM methodology, as well as the establishment of its application on mobile devices with telephone or written contact options, could facilitate an online and independent study conducted using CAMs with socio-culturally diverse populations. Possible differences regarding healthcare could then be investigated on a country-level and in more detail, including an in-depth content analysis of the concepts. In addition, CAMs could also be used in qualitative – for example interview – studies with socio-culturally diverse populations. Our study suggests that this could, for example facilitate the start of an interview and thus serve as a basis for a deeper exploration of a specific topic.

## Conclusion

Overall, the present exploratory study used a new measurement technique to gain qualitative insights into supporting and hindering factors in refugee healthcare. Despite experiencing multiple psychological challenges and facing various access barriers to healthcare, the participating refugees named numerous supporting factors that may help them cope with the experienced challenges. The findings indicated a variety of resources that refugees may utilise for themselves, as well as a preparedness to seek professional help in order to promote their well-being. The CAM measurement technique appears to be a promising tool for researching refugees' health needs by incorporating their expertise. This seems to enable a resource-oriented perspective and the recognition of the heterogeneity of the target group. Future larger-scale studies could build on these findings to enable a differentiated view when investigating refugee healthcare.

**Open peer review.** To view the open peer review materials for this article, please visit http://doi.org/10.1017/gmh.2025.10126.

**Supplementary material.** The supplementary material for this article can be found at http://doi.org/10.1017/gmh.2025.10126.

**Data availability statement.** The data that support the findings of this study are available from the corresponding author, KS, upon reasonable request.

**Acknowledgements.** The authors thank all study participants for their time and willingness to contribute.

**Author contributions.** LBS and KS initiated this study. EB, LBS, AK, IM, JF and KS contributed to the design of this study. Data acquisition was facilitated by AK, IM and JF by providing access and support to the CAM platform. EB largely contributed to the study preparation, recruitment, data collection, analyses and interpretation. IM and JF provided expertise in the use and analysis of CAM. AK, LBS and KS supervised the whole project. EB and KS wrote the draft of the manuscript. All authors contributed to the further writing of the manuscript and approved the final version of it.

**Financial support.** The authors acknowledge support by the Open Access Publication Fund of the University of Freiburg.

**Competing interests.** The authors declare none.

**Ethics statement.** The conduct of the study was approved by the ethics committee of the University of Freiburg (application number: 23–1548-S2).

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
