## [Reviewer Report]

This is a valuable study investigating the cultural differences in healthcare needs between various refugee groups in Germany through a novel technique-Cognitive-Affective Mapping (CAM). Incorporating such techniques into the research process could potentially improve engagement with culturally diverse populations. Please review my suggestions to improve the quality of the manuscript.

1. The introduction section could elaborate on the current state of the art concerning CAM. This would also strengthen the rationale of the study.

2. What are the research questions guiding the study?

3. Can authors describe the recruitment strategy further?

4. Since CAM is a novel technique, can the authors elaborate more on the exploratory findings? Participants` views on the technique and their (and researchers`) experiences with the process are important for future studies incorporating this technique.

5. Can authors discuss why the study participants had difficulties with the CAM technique (being different from other studies)?

6. Can authors make suggestions for future research? What are their suggestions for future researchers utilising this technique to engage with culturally diverse populations in the future?

---

## [Reviewer Report]

Dear Authors,

thank you very much for this very interesting paper. I enjoyed reading it. While I believe the paper is in good shape overall, I would still like to share some general comments that I hope will further strengthen the manuscript and support you in making minor revisions.

1) I noticed that the countries of origin of the refugees are not really discussed in the paper. I believe this information is highly relevant, as “West Africa” can refer to a number of countries, each with potentially different cultural contexts and lived experiences. Similarly, “Middle East” is not exact enough. Not specifying the countries of origin may create the same problem often seen in this area of research, where people from different countries are homogenized under broad regional labels. This overlooks possible (cultural) differences within these groups and limits the depth and nuance of the analysis. I recommend providing precise country-level information where possible (as you have done with “Ukraine”) to strengthen the insights and validity of the study. Have you found any nuances between groups on a country level? Also, why were these specific countries chosen?

2.) While I appreciate the differentiated way of presenting the results, I find it difficult to follow the statements about which group “named more challenges” than the other. Do you have any insights from your data into why this is the case? Was this observed consistently across all participants from the Middle East? (Referring to e.g., Lines 32–33 on Page 11.)

3) Reading your methods section, I wondered if you have discussed the language proficiency of the participants. Were interpreters present during the study? Were participants able to read, write, and speak German or English? How did the “joint collaboration” look in practice? Could this have had a potential impact on the participants’ responses?

4) Similar to my previous comment, on Page 12, Lines 8–11, I wonder if you can find evidence in the literature to support the finding and further elaborate why it is the case that Ukrainian refugees more frequently cited professional treatment compared to the other refugee populations. Also, while your conclusion that accessible and appropriate healthcare is needed is certainly true, I wonder if it might be equally important to consider ways to enhance knowledge of existing healthcare options and increase health literacy among these populations before focusing solely on access and treatment itself.

5) After reading your results section, I feel the discussion section could benefit from a more nuanced discussion of the central findings of your study. I would have appreciated a deeper exploration of how your findings relate to existing research. In particular, I missed a differentiated intra- and intergroup discussion of certain codes for the groups, and a clear comparison to the actual state of research is lacking. The sections comparing the different groups felt somewhat brief and descriptive, focusing primarily on quantitative numbers rather than offering an in-depth analysis of what was said, by whom, in comparison to whom, and why. Additionally, I think the results section would benefit from the inclusion of quotes, as you have provided them in the section on exploratory findings.

---

## [Reviewer Report]

Dear authors,

thank you for your efforts in maling changes to your manuscript. All points have been addressed. Congratulations on a very interesting paper!